# What are older smokers' attitudes to quitting and how are they managed in primary care? An analysis of the cross-sectional English Smoking Toolkit Study

Hannah Jordan,[1] Mira Hidajat,[2] Nick Payne,[1] Jean Adams,[3] Martin White,[3] Yoav Ben-Shlomo[2]

[1]Section of Public Health, School of Health and Related Research (ScHARR), The University of Sheffield, Sheffield, UK
[2]School of Social and Community Medicine, University of Bristol, Bristol, UK
[3]MRC Epidemiology Unit & CEDAR, School of Clinical Medicine, University of Cambridge, Institute of Metabolic Science, Cambridge Biomedical Campus, Cambridge, UK

**Correspondence to**
Dr Hannah Jordan;
h.c.jordan@sheffield.ac.uk

## ABSTRACT

**Objectives** To investigate whether age is associated with access to smoking cessation services.

**Design** Data from the Smoking Toolkit Study 2006–2015, a repeated multiwave cross-sectional household survey (n=181 157).

**Setting** England.

**Participants** Past-year smokers who participated in any of the 102 waves stratified into age groups.

**Outcome measures** Amount smoked and nicotine dependency, self-reported quit attempts and use of smoking cessation interventions. Self-report of whether the general practitioner (GP) raised the topic of smoking and made referrals for pharmacological support (prescription of nicotine replacement therapies (NRTs)) or other support (counselling or support groups).

**Results** Older smokers (75+ years) were less likely to report that they were attempting to quit smoking or seek help from a GP, despite being less nicotine-dependent. GPs raised smoking as a topic equally across all age groups, but smokers aged 70+ were more likely not to be referred for NRT or other support (ORs relative to 16–54 years; 70–74 years 1.27, 95% CI 1.03 to 1.55; 75–79 years 1.87, 95% CI 1.43 to 2.44; 80+ years 3.16, 95% CI 2.20 to 4.55; p value for trend <0.001).

**Conclusions** Our findings suggest that there are potential missed opportunities in facilitating smoking cessation in older smokers. In this large population-based study, older smokers appeared less interested in quitting and were less likely to be offered support, despite being less addicted to nicotine than younger smokers. It is unclear whether this constitutes inequitable access to services or reflects informed choices by older smokers and their GPs. Future research is needed to understand why older smokers and GPs do not pursue smoking cessation. Service provision should consider how best to reduce these variations, and a stronger effectiveness evidence base is required to support commissioning for this older population so that, where appropriate, older smokers are not missing out on smoking cessation therapies and the health benefits of cessation at older ages.

## INTRODUCTION

Despite a marked decline in smoking behaviour over the last few decades, around

---

**Strengths and limitations of this study**

► The findings of the study are based on a large, representative survey of the adult population of England.
► The large sample size enabled us to examine older age groups in far greater detail than in previous studies.
► Quit behaviours and support were self-reported, raising questions of differences in recall or reporting behaviour among older smokers.

---

10 million adults in the UK still smoke, of whom 11% (1.1 million) are over the age of 60.[1] There is, however, clear evidence for the benefits of quitting smoking at older ages. Large-scale prospective cohort studies have found that smokers who quit after the age of 65 years benefit from reduced mortality (2–4 extra years of life),[2] additional healthy life years[3] and reduced morbidity.[4] Although the relative risks associated with smoking status decline with age, the absolute risk differences continue to increase.[4]

The National Institute for Health and Care Excellence recommends that smoking cessation advice should be provided to all smokers, regardless of age. However current policy aims to target specific groups, such as pregnant women and socioeconomically disadvantaged populations for cessation advice, while older smokers are not generally recognised as a priority group.[5] A recent study reported that opportunities to offer cessation advice and support to older smokers may often be missed in primary care.[6] However, a recent review showed that pharmacotherapy such as nicotine replacement therapy (NRT) was effective in older adults, but that the literature was sparse and effective smoking

cessation services may need to be tailored to the needs of older adults.[7]

There are several possible reasons why smoking cessation therapies may not be used equally across age groups. It has been suggested that older smokers may be more strongly addicted to nicotine, which may hamper attempts to initiate quitting.[8] Older people themselves may have beliefs about quitting, considering themselves 'survivors' or believing that 'the damage is done' so they see no point in attempting to quit later in life, resulting in reluctance to demand services or recognise the benefits of quitting.[9] Health professionals may have beliefs about older smokers that hinder their access to smoking cessation services, such as a reluctance to give cessation advice or to provide medication.[10–13] The type, location and visibility of smoking cessation services are decided locally,[14] and may introduce barriers by age. For example, older smokers may be reluctant to use telephone or online support such as text messaging.[15]

In this paper, we examine the potential reasons why older smokers may or may not have equitable access to smoking cessation therapies in a large, multiwave, national, cross-sectional survey in England. Because of the large sample size, we were able examine older age groups in far greater detail than in previous studies, using 5-year age bands up to age 80+ rather than aggregating data for those over 65 or even younger. We compare older smokers with younger smokers to test if they are as likely to (1) attempt to quit or cut down on their cigarette consumption, (2) be nicotine-dependent and (3) seek help from a general practitioner (GP). In addition, we test whether GPs are as likely, when comparing older with younger smokers, to (4) discuss smoking cessation and (5) refer patients to either smoking cessation services and/or prescribe NRTs.

## METHODS

This manuscript was written in adherence to the Strengthening the Reporting of Observational Studies in Epidemiology statement.[16]

### Data

We used data from the Smoking Toolkit Study (STS), a repeated, multiwave, cross-sectional household survey conducted approximately every month from November 2006 to January 2015 across England (102 waves of data, n=181 157).[15] Each survey wave contains responses from approximately 1800 adults aged 16+ (approximately 570 ex-smokers and 500 current smokers) collected during a face-to-face computer-assisted household survey. The sampling method is a hybrid between random probability and simple quota. The sample is identified by a random selection of over 170 000 localities ('output areas') after stratification by a geodemographic analysis of the population. Each locality contains approximately 250 households. Interviewers visit households within the locality starting at a random point in the area. One member per household is interviewed until interviewers achieve local quotas designed to minimise differences in the probability of participation. The method has been shown to result in a sample that is nationally representative in its sociodemographic composition. Although the data are not publicly available, they can be requested from the STS.[17]

### Exposure

Our exposure of interest was age. We chose to operationalise age into bands, not assuming any linear relationship, as follows: 16–54 years as baseline, then 5-year bands from 55 to 79, and 80+ as the oldest age group. Age bands were selected to allow us to look in detail at older smokers' experience, while retaining sufficient numbers in each band for analysis.

### Smoking status, nicotine dependency and reducing consumption

We classified smoking status using the four categories in the STS: current smoker (respondents who indicated that they smoke cigarettes (including hand-rolled) or other forms of tobacco (eg, pipe or cigar); recent smoker (respondents who indicated that they have stopped smoking within the past year); past smoker (respondents who indicated that they have not smoked for a year or more); and never smokers. For current, recent and past smokers, we categorised respondents into groups by the amount smoked: 10 or fewer cigarettes per day, 11–20 cigarettes per day, and 21 or more cigarettes per day.

### Dependency and past use of services

Nicotine dependency (yes/no) was classified using the Heaviness of Smoking Index,[18] a validated index based on a subset of questions from the Fagerstrom Test of Nicotine Dependence[19] (see box for more details concerning specific outcomes).

We defined smoking cessation and reduction behaviours—quit attempts and cutting down in the past 12 months—as dichotomous variables due to small numbers of respondents with more than one quit attempt. We classified whether current smokers used two types of smoking cessation interventions (yes/no)—use of NRTs and NHS Stop Smoking counselling.

### GP–patient interaction and management

The survey captures both patient-initiated and GP-initiated interactions around smoking cessation, allowing us to examine whether relevant respondents raised the topic of smoking with their GPs and also whether the doctor raised the issue of smoking. For the latter question there were a series of outcome options, such as a referral for stop smoking counselling, prescription of an NRT, both, or advice but no referral. To look at the determinants of advice only, we created a dichotomous variable by comparing no referral for counselling, prescription of NRT or both. Finally there was an option that the GP did not specifically advise the patient to stop smoking (no active management).

---

**Box    Definitions of terms and outcomes**

**Nicotine dependency** (yes/no): derived from Heaviness of Smoking Index combining two scales (0–3) on 'How soon after you wake up do you smoke your first cigarette?' (sooner gets higher score) and 'How many cigarettes per day do you usually smoke?' (more cigarettes gets higher score). Scores from both questions are added for each respondent (min=0, max=6), and we dichotomised this variable into a high (3–6) and low (0–2) score.

**Quit attempt** (yes/no): this was derived from the question 'How many serious attempts to stop smoking have you made in the last 12 months?' asked to current smokers and recent smokers (in the last 12 months). This was dichotomised as any serious attempts to stop smoking within the last year or no attempts.

**Cutting down** (yes/no): this was only asked of current smokers and was coded from the question 'Are you trying to cut down on how much you smoke?' (waves 1–32), which was later changed to 'Are you currently trying to cut down on how much you smoke but not currently trying to stop?' (waves 33–66, 68, 70, 72, 74, 76, 78–102).

**Use of NRTs** (yes/no): respondents who were current smokers trying to cut down could report use of one or more of the following NRT products: nicotine replacement gum, lozenges/tablets, inhaler, nasal spray, patch or mouth spray. We dichotomised use of any NRT products versus none (question not included in waves 67, 69, 71, 73, 75 and 77).

**Use of NHS advice services** (yes/no): current smokers who responded that they had at least one serious quit attempt were shown a list of potential cessation services: NHS Stop Smoking Service group; one-to-one counselling/advice support session; contact with local NHS Stop Smoking Service; NHS smoking helpline; or Quitline. Use of NHS Smokefree website was not included in this list because it was available to a small subset of the survey (waves 26–102).

**Patient raised smoking** (yes/no): this was captured by a dichotomous variable to the question 'Have you raised the topic of smoking with your GP in the past year (ie, last 12 months)?' asked of current smokers and recent ex-smokers (quit within the last 12 months).

**Doctor raised smoking** (yes/no): this came from the question 'Has your GP spoken to you about smoking in the past year (ie, last 12 months)?' This question was asked in waves 31–39 to current and former smokers (quit within the last 12 months or more) and in waves 40–102 to current smokers and recent ex-smokers (quit within the last 12 months).

**Doctor management**: if the doctor raised smoking was YES, then the following responses were available: (1) he/she suggested that I go to a specialist stop smoking advisor or group or that I see a nurse in the practice (**counselling**); (2) he/she offered me a prescription for a nicotine patch, nicotine gum or other nicotine product (**NRT prescription**); (3) he/she suggested that I go to a specialist stop smoking advisor or group, or see a nurse in the practice and offered me a prescription for a nicotine patch, nicotine gum or another nicotine product (**counselling and NRT prescription**); (4) yes, he/she advised me to stop but did not offer anything (**advice only**); (5) yes, he/she asked me about my smoking but did not advise me to stop smoking (**no active management**). These responses are available in waves 40–102 to current smokers and recent ex-smokers (quit within the last 12 months).

GP, general practitioner; NHS, National Health Service; NRT, nicotine replacement therapy.

---

## Potential confounders

We examined the following potential confounders: gender, secular period as measured by year of survey (2-year bands) and a measure of socioeconomic status ('social grade'—see below). We argued that both social grade and gender could have influenced past smoking behaviour and may determine willingness to engage in cessation therapies, as well as being age-related (more women and more affluent older participants). Secular period may influence availability of cessation therapies such as counselling as services are reprocured and reconfigured over time.

Social grade was reported using the five occupation-based categories of the National Readership Survey method: AB (higher/intermediate managerial, administrative or professional), C1 (supervisory, clerical and junior managerial, administrative and professional), C2 (skilled manual workers), D (semiskilled and unskilled manual workers) and E (state pensioners, casual and lowest grade workers, unemployed with state benefits only[20]). The social grade of the household head was used to classify the entire household.

## Statistical analyses

We conducted logistic regression analyses based on the relevant outcome variables (nicotine dependency, quit attempts, cutting down, past NRT usage, past National Health Service (NHS) advice services used, patient or GP raised smoking, GP prescription of NRT, stop smoking counselling, or just advice). Independent variables were age, gender, social grade and survey year. We modelled age group both as a dummy variable and as an ordinal variable to test for a linear trend. Ordered logistic regression was performed for categories of amount smoked per day. To examine how management differed by age group, we derived the ORs (95% CI, p value), using logistic regression models, for referral for counselling, NRT prescription, both or no active management, as compared with the baseline group of the GP giving advice to quit (advice only) but no referral. For each model we ran unadjusted and adjusted models conditioning on gender, social grade and secular period. Due to questionnaire changes, some questions were not available in all waves, particularly doctor–patient interactions, NHS Stop Smoking Service use and NRT use. In our analyses, we used the appropriate waves for each model accordingly, although it meant using different waves of data across the various models (see box). We conducted a sensitivity analysis using only the survey waves with complete data for GP-related questions (waves 72–102).

## RESULTS

### Descriptive analyses

From a total sample of 181 157 participants, there were 41 031 (22.7%) current smokers, 2825 (1.6%) recent

**Table 1** Association between sociodemographics and time period with smoking status and number of cigarettes consumed per day

| | Smoking status | | | | | | Cigarettes per day | | |
| | Current versus never smoker | | | Recent/past versus never smoker | | | | | |
| | OR | p Value | 95% CI | OR | p Value | 95% CI | OR* | p Value | 95% CI |
|---|---|---|---|---|---|---|---|---|---|
| **Age** | | | | | | | | | |
| 16–54 | 1.00 | | | 1.00 | | | 1.00 | | |
| 55–59 | 0.92 | <0.001 | 0.88 to 0.97 | 1.79 | <0.001 | 1.70 to 1.89 | 1.80 | <0.001 | 1.68 to 1.93 |
| 60–64 | 0.71 | <0.001 | 0.68 to 0.75 | 2.19 | <0.001 | 2.09 to 2.29 | 1.91 | <0.001 | 1.79 to 2.04 |
| 65–69 | 0.62 | <0.001 | 0.59 to 0.66 | 2.65 | <0.001 | 2.53 to 2.78 | 1.79 | <0.001 | 1.66 to 1.92 |
| 70–74 | 0.42 | <0.001 | 0.39 to 0.44 | 2.41 | <0.001 | 2.30 to 2.53 | 1.51 | <0.001 | 1.39 to 1.64 |
| 75–79 | 0.29 | <0.001 | 0.27 to 0.31 | 2.35 | <0.001 | 2.23 to 2.48 | 1.18 | 0.001 | 1.07 to 1.30 |
| 80–84 | 0.21 | <0.001 | 0.19 to 0.23 | 2.44 | <0.001 | 2.29 to 2.60 | 0.94 | 0.330 | 0.82 to 1.07 |
| 85+ | 0.13 | <0.001 | 0.12 to 0.15 | 2.36 | <0.001 | 2.19 to 2.54 | 0.73 | 0.001 | 0.61 to 0.89 |
| p Value for trend | <0.001 | | | <0.001 | | | <0.001 | | |
| **Gender** | | | | | | | | | |
| Men | 1.34 | <0.001 | 1.31 to 1.37 | 1.39 | <0.001 | 1.35 to 1.43 | 1.49 | <0.000 | 1.44 to 1.54 |
| Women | 1.00 | | | 1.00 | | | 1.00 | | |
| **Social grade** | | | | | | | | | |
| AB | 1.00 | | | 1.00 | | | 1.00 | | |
| C1 | 1.56 | <0.001 | 1.50 to 1.63 | 1.00 | 0.897 | 0.97 to 1.04 | 1.03 | 0.416 | 0.96 to 1.09 |
| C2 | 2.40 | <0.001 | 2.31 to 2.50 | 1.09 | <0.001 | 1.05 to 1.14 | 1.31 | <0.001 | 1.23 to 1.39 |
| D | 2.88 | <0.001 | 2.76 to 3.00 | 0.97 | 0.156 | 0.92 to 1.01 | 1.31 | <0.001 | 1.23 to 1.40 |
| E | 4.72 | <0.001 | 4.52 to 4.92 | 0.96 | 0.078 | 0.92 to 1.00 | 1.62 | <0.001 | 1.53 to 1.72 |
| p Value for trend | <0.001 | | | 0.008 | | | <0.001 | | |
| **Survey years** | | | | | | | | | |
| 2006–2008 | 1.00 | | | 1.00 | | | 1.00 | | |
| 2009–2010 | 0.88 | <0.001 | 0.86 to 0.91 | 0.90 | <0.001 | 0.86 to 0.94 | 0.86 | <0.001 | 0.82 to 0.90 |
| 2011–2012 | 0.88 | <0.001 | 0.86 to 0.91 | 1.14 | <0.001 | 1.10 to 1.19 | 0.64 | <0.001 | 0.61 to 0.68 |
| 2013–2015 | 0.80 | <0.001 | 0.77 to 0.83 | 1.26 | <0.001 | 1.22 to 1.31 | 0.55 | <0.001 | 0.52 to 0.58 |
| p Value for trend | <0.001 | | | <0.001 | | | <0.001 | | |

*Ordered logistic regression with the following outcomes: 10 or fewer cigarettes per day, 11–20 cigarettes per day, and 21 or more cigarettes per day.

smokers, 25 367 (14.0%) past smokers and 112 046 (61.9%) never smokers and 131 missing (0.07%). Older participants were less likely to be current smokers and more likely to be former smokers, and there was an inverted J-shaped relationship for smoking intensity, with those aged 55–59 years old and those >65 years old less likely to be heavy smokers (table 1). Online supplementary table 1 describes each outcome by sociodemographic characteristics and survey year. The numbers of participants included in each analysis depend on the numbers of respondents for individual survey questions.

### Nicotine dependency and quit attempts
Nicotine dependency data were available for 51 920 people. Dependency showed an inverted J pattern so that participants aged 55–69 appeared to have higher

nicotine dependency, but from 70 years upwards there was a progressive reduction in the OR for dependency (table 2). However, older participants from 55 years onwards were less likely to report having at least one quit attempt. (Compared with those aged 16–54 years old, the OR for quit attempts in the past year for those aged 55–59 years old was 0.74 (0.68 to 0.80), reducing to 0.32 (0.22 to 0.46) for those aged 85+.) Men showed greater dependency and were less likely to quit. The odds of high nicotine dependency increased from higher to lower social grades, and respondents from the lower social grades were also less likely to report a recent quit attempt. Participants from surveys in more recent years were less dependent and less likely to make quit attempts.

**Table 2** Association between sociodemographics and time period with nicotine dependency and quit attempts over the past year

| | High nicotine dependency | | | Any quit attempts in the past year | | |
|---|---|---|---|---|---|---|
| | OR | p Value | 95% CI | OR | p Value | 95% CI |
| **Age** | | | | | | |
| 16–54 | 1.00 | | | 1.00 | | |
| 55–59 | 1.35 | <0.001 | 1.26 to 1.44 | 0.74 | <0.001 | 0.68 to 0.80 |
| 60–64 | 1.31 | <0.001 | 1.22 to 1.40 | 0.70 | <0.001 | 0.64 to 0.76 |
| 65–69 | 1.17 | <0.001 | 1.08 to 1.26 | 0.65 | <0.001 | 0.59 to 0.71 |
| 70–74 | 0.98 | 0.669 | 0.90 to 1.07 | 0.57 | <0.001 | 0.51 to 0.64 |
| 75–79 | 0.69 | <0.001 | 0.62 to 0.76 | 0.41 | <0.001 | 0.35 to 0.48 |
| 80–84 | 0.54 | <0.001 | 0.47 to 0.63 | 0.27 | <0.001 | 0.21 to 0.35 |
| 85+ | 0.45 | <0.001 | 0.37 to 0.56 | 0.32 | <0.001 | 0.22 to 0.46 |
| p Value for trend | | <0.001 | | | <0.001 | |
| **Gender** | | | | | | |
| Men | 1.21 | <0.001 | 1.17 to 1.26 | 0.81 | <0.001 | 0.78 to 0.85 |
| Women | 1.00 | | | 1.00 | | |
| **Social grade** | | | | | | |
| AB | 1.00 | | | 1.00 | | |
| C1 | 1.20 | <0.001 | 1.12 to 1.28 | 0.96 | 0.23 | 0.89 to 1.03 |
| C2 | 1.61 | <0.001 | 1.51 to 1.72 | 0.91 | 0.01 | 0.85 to 0.98 |
| D | 1.89 | <0.001 | 1.77 to 2.02 | 0.88 | 0.001 | 0.81 to 0.95 |
| E | 2.44 | <0.001 | 2.29 to 2.61 | 0.90 | 0.005 | 0.84 to 0.97 |
| p Value for trend | | <0.001 | | | <0.001 | |
| **Survey years** | | | | | | |
| 2006–2008 | 1.00 | | | 1.00 | | |
| 2009–2010 | 0.95 | 0.03 | 0.91 to 0.97 | 0.80 | <0.001 | 0.76 to 0.84 |
| 2011–2012 | 0.80 | <0.001 | 0.76 to 0.85 | 0.73 | <0.001 | 0.69 to 0.77 |
| 2013–2015 | 0.72 | <0.001 | 0.69 to 0.76 | 0.85 | <0.001 | 0.81 to 0.90 |
| p Value for trend | | <0.001 | | | <0.001 | |

### Past use of NRT or stop smoking counselling

Past use of NRTs was calculated on data from 20 286 participants. Use was more common among those aged 55–64 years old compared with those aged 16–54 years old (OR 1.13 (1.03 to 1.25)); use of stop smoking counselling (16 026 participants) was more common among those aged 65–74 years old (OR 1.38 (1.09 to 1.74)). Older participants (75+ years) were less likely to have been prescribed NRT than those aged 55–64 or 65–74 years old (figure 1; OR 0.70). Data on participants over the age of 75 were combined due to small numbers in the oldest age groups. We found similar patterns by gender; men were less likely than women to have been prescribed NRTs or referred for counselling (see online supplementary table 2). Participants in social grade D were less likely to have been prescribed NRTs than those in social grade AB, but there were no social grade differences in referral for counselling. There appeared to be a secular pattern so that respondents from recent survey years (2011–2015) were less likely to have been prescribed NRTs and referred for counselling than those from 2006 to 2008.

### Raising smoking in consultations and management

We found an inverted J-shaped relationship for age group, such that older adults (75 years and above) were much less likely to have raised the topic of smoking during their GP consultation (OR for 75–79 years compared with 16–54 years of 0.49 (0.31 to 0.78). In contrast GPs were more likely to have raised smoking for all age groups compared with younger smokers except the 80+ age group, where there was no evidence of any difference (table 3).

However, smokers over 70 years were less likely to report that their GPs offered any support to help quitting.

Despite discussing smoking with all age groups, GPs demonstrated marked differences in how they managed smokers. The self-reported probability of being prescribed NRT fell with increasing age (after 75 years) as did referral for counselling (after 70 years). Similar patterns

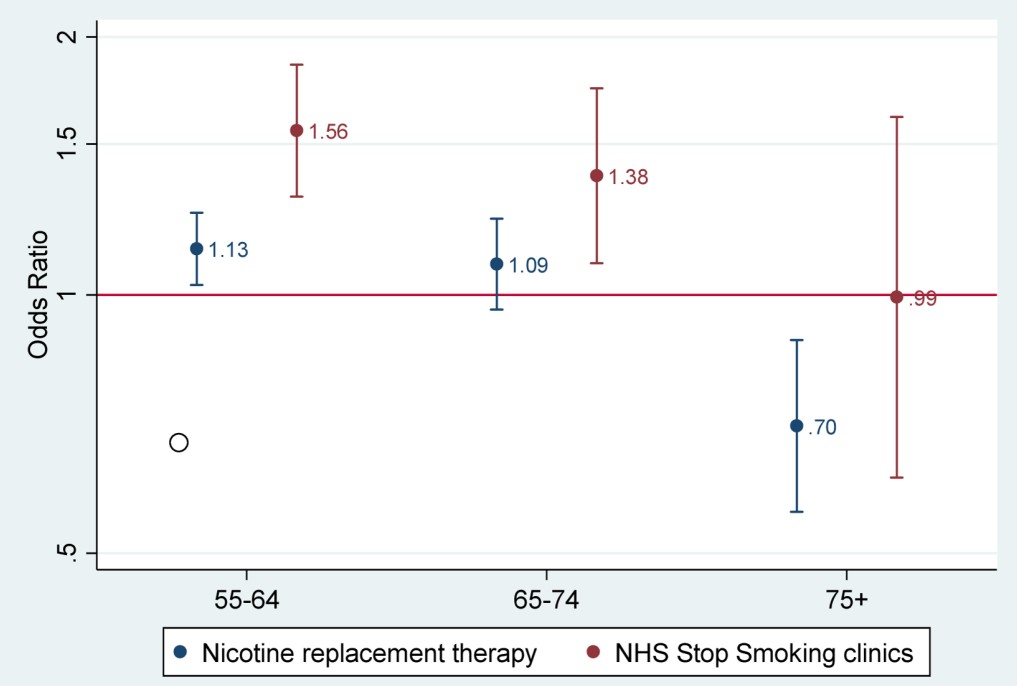

**Figure 1** ORs for past use of nicotine replacement therapies and NHS Stop Smoking clinics by age group. Age reference is 16–54 years. Models are adjusted for gender, social grade and survey year. NHS, National Health Service.

were seen with prescription of NRT plus referral for counselling, but these estimates were imprecise due to small numbers. The OR for not being advised to quit smoking was elevated after 65 years though for the 80+ age group: this was consistent with chance variation (figure 2; see online supplementary table 3).

There was a marked gender difference—reports from male respondents indicate that they and their GPs were less likely to raise smoking during consultation. Men also reported being given less support to quit smoking. A concordant pattern was also observed for social grade such that lower grade was associated with both patients and doctors raising smoking and modest evidence that they received more support than higher grades. GPs appeared to be less likely to raise smoking with male patients and in more recent survey years.

In comparison to our original models, most of the results from the sensitivity analyses remained unchanged, the main difference being loss of statistical significance due to reduced power (see online supplementary table 4).

## DISCUSSION
Our findings suggest that there are potential missed opportunities in facilitating smoking cessation in older smokers. We observed that older smokers, despite being less dependent on nicotine and smoking fewer cigarettes, key predictors of success in quitting,[21] were also less likely to have attempted to quit. This was also partially reflected in less past experience with NRTs or referral for counselling. This may reflect reduced motivation as older

smokers were less likely to raise smoking as a topic in their GP consultations, but we did note that participants in the period around retirement were more likely to raise this. Transitional events such as retirement have been linked to success in quitting.[22] The relatively high success rate in quitting in this age group has been noted in the literature,[20] but evidence is still limited.[23] GPs appeared generally to be equitable in raising smoking as a topic across all age groups, but older participants were less likely to report being supported to quit. This was reflected in fewer prescriptions of NRTs and fewer referrals for counselling.

We cannot be certain whether these results demonstrate genuine inequities in receiving smoking cessation services for older smokers or reflect patient preferences which may be known to their GP, hence no attempt to offer specific interventions. Previous studies have reported that medical professionals are less likely to advise older patients to stop smoking in a hospital setting[9] and less likely to prescribe NRT or other smoking cessation medications to those over 60 years,[24 25] consistent with our observations.

Past research indicates that some older adults have low motivation to quit, believing either that 'the damage is done' and that quit attempts are unlikely to be successful after a lifetime of smoking, or that the harm of smoking is exaggerated or does not apply to them.[21 24 25] However, 'trigger events' such as an episode of ill health can still result in quit attempts, as can prompts from family and health professionals.[9] NRT has also been viewed with suspicion by some older smokers,[9] but uptake can be high,[26] and recent reviews of the evidence for effectiveness in older

**Table 3** Association between sociodemographics and time period with discussion of smoking at general practitioner consultation and whether the doctor offered any active management

| | Patient raised smoking | | | Doctor raised smoking | | | Doctor no support | | |
|---|---|---|---|---|---|---|---|---|---|
| | OR | p Value | 95% CI | OR | p Value | 95% CI | OR | p Value | 95% CI |
| Age | | | | | | | | | |
| 16–54 | 1.00 | | | 1.00 | | | 1.00 | | |
| 55–59 | 1.31 | 0.006 | 1.08 to 1.58 | 1.58 | <0.001 | 1.44 to 1.74 | 1.14 | 0.126 | 0.97 to1.34 |
| 60–64 | 1.47 | <0.001 | 1.22 to 1.78 | 1.88 | <0.001 | 1.72 to 2.07 | 1.04 | 0.636 | 0.89 to 1.22 |
| 65–69 | 1.05 | 0.666 | 0.84 to 1.31 | 1.98 | <0.001 | 1.78 to 2.19 | 0.90 | 0.233 | 0.75 to 1.07 |
| 70–74 | 0.96 | 0.774 | 0.73 to 1.27 | 1.85 | <0.001 | 1.63 to 2.09 | 1.27 | 0.023 | 1.03 to 1.55 |
| 75–79 | 0.49 | 0.002 | 0.31 to 0.78 | 1.69 | <0.001 | 1.44 to 1.98 | 1.87 | <0.001 | 1.43 to 2.44 |
| 80+ | 0.48 | 0.005 | 0.28 to 0.80 | 1.02 | 0.804 | 0.85 to 1.24 | 3.16 | <0.001 | 2.20 to 4.55 |
| p Value for trend | | <0.001 | | | <0.001 | | | <0.001 | |
| Gender | | | | | | | | | |
| Men | 0.81 | <0.001 | 0.74 to 0.90 | 1.72 | <0.001 | 0.69 to 0.76 | 1.15 | 0.002 | 1.05 to 1.26 |
| Women | 1.00 | | | 1.00 | | | 1.00 | | |
| Social grade | | | | | | | | | |
| AB | 1.00 | | | 1.00 | | | 1.00 | | |
| C1 | 1.12 | 0.259 | 0.92 to 1.36 | 1.02 | 0.724 | 0.93 to 1.11 | 0.96 | 0.614 | 0.81 to 1.13 |
| C2 | 1.01 | 0.897 | 0.83 to 1.24 | 1.06 | 0.190 | 0.97 to 1.16 | 0.87 | 0.097 | 0.73 to 1.03 |
| D | 1.24 | 0.036 | 1.02 to 1.51 | 1.09 | 0.073 | 0.99 to 1.19 | 0.90 | 0.210 | 0.76 to 1.06 |
| E | 1.70 | <0.001 | 1.40 to 2.07 | 1.47 | <0.001 | 1.35 to 1.60 | 0.83 | 0.023 | 0.70 to 0.97 |
| p Value for trend | | <0.001 | | | <0.001 | | | 0.017 | |
| Survey years | | | | | | | | | |
| 2009–2010 | | | | 1.00 | | | 1.00 | | |
| 2011–2012 | | | | 0.88 | <0.001 | 0.83 to 0.93 | 0.99 | 0.93 | 0.89 to 1.12 |
| 2013–2015 | | | | 0.79 | <0.001 | 0.75 to 0.84 | 1.18 | 0.01 | 1.05 to 1.32 |
| p Value for trend | | | | | <0.001 | | | <0.001 | |
| 2012–2013 | 1.00 | | | | | | | | |
| 2014–2015 | 0.85 | 0.003 | 0.77 to 0.95 | | | | | | |

people showed that, although there is limited research in the oldest age groups, NRT appears to be effective in adults over 60 years.[7 27] Although the benefits of smoking cessation for life expectancy accumulate with years since quitting, and there is still relatively little evidence on the benefits to people over the age of 65, it has been argued that smoking-related risks such as myocardial infarction and vascular disease are not modified by age,[28] and that healthy life years may be important to emphasise for an older population.[3]

We also noted important gender differences such that men were less likely to quit or cut down, less likely to use NRTs or any NHS cessation services, and less likely to speak to their GPs about smoking. Some of these may reflect their greater level of nicotine dependency. This does not explain, however, why GPs were reportedly less likely to raise smoking with men and less likely to offer any form of support. More reassuringly, while participants of lower social grade were more nicotine-dependent, they were more likely to speak to their GPs about smoking, and similarly GPs were more likely to raise this as a topic. We found no

obvious social grade differences in GP referrals for counselling or prescription of NRTs. We had originally postulated 'a priori' that it might be harder to get more recent cohorts to quit as the prevalent pool of smokers might be enriched with heavy smokers who find it difficult to quit. In fact, the time trends suggested the opposite pattern, as dependency is less in more recent cohorts.

### Strengths and limitations
Because of the large sample size, we were able examine older age groups in far greater detail than in previous studies, allowing greater insights into patient and GP behaviours. Previous studies on stopping smoking have typically aggregated all older age groups into one category of age 65 and older, even though this open-ended category is composed of many cohorts with different smoking initiation and cessation behaviours.[7 12 29–31]

Second, we were able to estimate socioeconomic status using an occupation-based indicator, which previous research has found to be one of the strongest

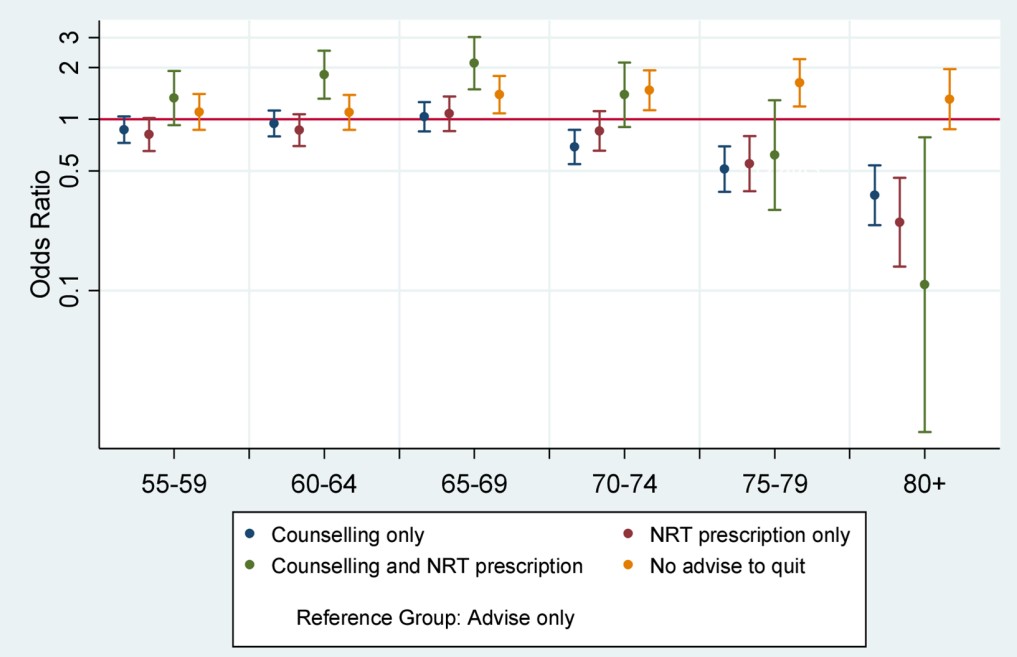

**Figure 2** ORs for general practitioner management (nicotine replacement therapies (NRTs), counselling, both or no advice to quit) of smokers by age groups. Age reference is 16–54 years. Models are adjusted for gender, social grade and survey year.

socioeconomic status measures, particularly for examining health issues in elderly populations in the UK.[32] Nevertheless, this measure captures only one dimension of socioeconomic status. A more multidimensional measure may have further reduced any residual confounding by socioeconomic status, but without postcode data we were not able to add this.

The STS captures self-reported information on smoking behaviour and experience. These measures record patients' perceptions on interactions or lack thereof with their GPs and are not based on direct observations or records of doctor–patient interactions. If older patients tend to recall or report their experiences differently from younger patients, the findings may be biased. For example, if older patients fail to recall GP advice, this would exaggerate our observation that GPs provided less support for older participants. Second, our data are from an English population, so it is unclear whether our findings would be generalisable to the rest of the UK. The STS does not provide us data on comorbidity or have any follow-up data on mortality. It is therefore possible that GPs choose not to try to change smoking behaviours in those patients who have multiple morbidities or a poor life expectancy, which may well be clinically appropriate.

Finally, our analyses are based on respondents. While we have no data on non-response bias, it is reasonable to assume that those who chose not to participate in the voluntary STS are more likely to be older and that those who did respond are probably healthier than the general older population. While this may limit generalisability, it is likely that this subsample is of greater clinical interest as they are more likely to benefit from stopping smoking than frailer individuals with potentially more limited life expectancy.

### Implications for research, policy and practice
It is unclear from our findings what proportion of smokers who were not offered interventions to stop smoking may have benefited from them and would have wished to consider these options. Clearly smokers are under no obligation to try and quit, but we suspect in some cases this may be due to not understanding the potential benefits or assuming there are none. Qualitative research could help to gain greater insight into why older patients and their GPs do not pursue smoking cessation, so that appropriate interventions can be designed to reduce inequitable access. For example, recent research has demonstrated that the use of targeted invitations to stop smoking services increased attendance in all age groups, including people aged 65+, with the relative benefits larger in this age band than younger groups.[33] This supports our view that it may be easier to change behaviour in this group if motivated. Older adults need to be offered smoking cessation interventions that are acceptable, appropriate and available, even if they decide not to use them. With the recommissioning of smoking cessation services, there is an opportunity to ensure that currently underserved groups are supported in attempts to stop smoking or reducing their intake. Evaluative studies are required to determine the most cost-effective type, location and mode of delivery of these interventions to ensure that older smokers can also realise the health benefits of quitting into their 60s and beyond.

### CONCLUSIONS
This large national study has demonstrated that older smokers report being less likely to quit or reduce consumption, although are less nicotine-dependent. Potential

opportunities to facilitate cessation are being missed as older smokers report that GPs are less likely to offer interventions or specific advice, although this may reflect past patient preferences. Service provision should consider how best to reduce these variations, and a stronger effectiveness evidence base is required to support commissioning for this older population.

**Acknowledgements** We thank Jamie Brown and Robert West of the Clinical, Educational and Health Psychology, Division of Psychology and Language Sciences, Faculty of Brain Sciences, University College London, for their support with accessing the Smoking Toolkit Study.

**Contributors** The study was initially conceived and designed by YB-S, HJ and MH, with contributions from all authors. Data analysis was carried out by MH and YB-S. All authors contributed to drafting and critically revising the paper. All authors have approved the final version and are equally accountable for the work.

**Funding** The Smoking Toolkit Study is currently primarily funded by Cancer Research UK (C1417/A14135; C36048/A11654; C44576/A19501), and has previously also been funded by Pfizer, GlaxoSmithKline and the Department of Health. MW and JA are funded by the Centre for Diet and Activity Research (CEDAR), a UKCRC Public Health Research Centre of Excellence which is funded by the British Heart Foundation, Cancer Research UK, Economic and Social Research Council, Medical Research Council, the National Institute for Health Research, and the Wellcome Trust. This work was undertaken as part of the Ageing Well Programme, funded by the NIHR School for Public Health Research (SPHR). The views expressed are those of the author(s) and not necessarily those of the NHS, the NIHR or the Department of Health.

**Competing interests** None declared.

**Ethics approval** Ethical approval was granted to the STS, and the anonymised study data were used under licence.

**Provenance and peer review** Not commissioned; externally peer reviewed.

**Data sharing statement** The data that support these findings are available from the Smoking Toolkit Study. Restrictions apply to the availability of these data, which were used under licence for the current study, and so are not publicly available. Data are however available from the authors on reasonable request and with the permission of Robert West at robertwest100@googlemail.com.

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
