## [Reviewer comments · BMJ Open]

ARTICLE DETAILS

TITLE (PROVISIONAL)	What are older smokers' attitudes to quitting and how are they managed in primary care? An analysis of the cross-sectional English Smoking Toolkit Study
AUTHORS	Jordan, Hannah; Hidajat, Mira; Payne, Nick; Adams, J; White, Martin; Ben-Shlomo, Yoav

VERSION 1 – REVIEW

REVIEWER	George Tsourtos Flinders University, Australia
REVIEW RETURNED	23-Jun-2017

GENERAL COMMENTS	Well written and presented. Good to see research on the less sexy topic of older smokers, at a time when we have ageing populations. However, this is a point that the authors could include in the paper. Method: Could include justification as to how you defined past smoker on the basis of 1 year, as well as the age bands selected. Make clear earlier (under heading - 'dependency and past use of services') why you dichotomised reduction behaviour variables. Explicitly state which variables you entered as independent variables in the statistical analysis. If there is any background on the GPs include this as well - for example, GPs gender, age, if sole practitioner, etc Implications: there is an appropriate suggestion of the need to do qualitative research. However, there is not a great deal of discussion of implications for policy and practice in direct relation to the authors' data. Conclusion: is not saying much - it is mostly just a summary. There are a few sentences that overuse comas. For example, in line 54 in the abstract Overall, this is an interesting an important study. The negative points I have raised are mostly to do with the end of the paper but they can easily be remedied.
---

REVIEWER	Kei Long Cheung Maastricht University, The Netherlands
REVIEW RETURNED	12-Jul-2017

GENERAL COMMENTS	The manuscript used data from the Smoking Toolkit Study and showed some insights in the relations between age, smoking, and cessation services. The paper is interesting and well-written with proper data and analyses. Yet, I have several comments.  • In the abstract, the objective, I wonder whether 'influenced' is the correct term. Please reconsider. • In the first sentence of the introduction, I think relative numbers of the smoking problem would make it more comprehensible for the reader. • In the second paragraph of the introduction, line 20-21, it is mentioned that smoking cessation advice should be provided to all smokers regardless of age including the oldest old. I am wondering what the arguments were. I suggest stating these. Also, it is not clear what is meant with oldest old. • "Advice and support to older smokers may often be missed..." (line 28), I am wondering whether there are any estimates showing how often. • Line 30-31, I miss some elaboration about the potential positive effects of other interventions than NRT. • In the last paragraph of the introduction, line 8, it is not entirely clear what is meant with "far greater detail". • In the same paragraph and line, authors mentioned several hypotheses while they look like research questions to me. I suggest rephrasing them. • In the methods, line 51, I am wondering whether there were specific reasons for using 5 year bands and 80+ as oldest age-group. If there are reasons, the paper would benefit from more transparency. • Page 8, line 3-4, it is not clear how smoking was defined (e.g. not a single puff?). • Page 8, line 10-13, I am wondering whether there is a reference supporting this classification. For instance, the 21 or more cigarettes per day would be classified as heavy smokers? • Page 8, line 25, is there an argument that quit attempts is a good outcome measure for this study? For instance, smoking cessation could also be defined as prolonged abstinence of at least 6 months or even continuous abstinence. Also I suggest adding "in the past 12 months" to clarify what this quit attempts captures. • In the results section, last paragraph, the results from the sensitivity analyses are not reported properly. I think the paper would benefit from more details. I suggest adding this information in a supplementary table.
---

	 • Page 12, line 11-16, it is discussed that previous studies reported that medical professionals are less likely to advise older patients to stop smoking in a hospital setting and less likely to prescribe NRT or other smoking cessation medications to those over 60 years of age. I am wondering what the reasons are. • In the next paragraph, line 20, the low motivation to quit is discussed. I miss some link of findings to motivational interventions that may be provided to this group. What is the role of motivational interviewing for instance? • Page 13, line 40, "this is greater for... healthier than the general older population". I am wondering whether there is support for this direction of non-response bias. • It seems to me that this is already described elsewhere but I am wondering whether this paper should report on the research ethics too.
--	---

REVIEWER	David Blane University of Glasgow, UK
REVIEW RETURNED	17-Jul-2017

GENERAL COMMENTS	This paper uses data from a large cross-sectional household survey study (STS) of self-reported smoking behaviour and treatment in England. It is generally well written, but despite this I found it difficult to follow. This is partly due to the large number of outcome variables (smoking status, cigarettes per day, nicotine dependency, quit attempts in past year, cutting down, use of NRT, use of NHS advice services, pt raised smoking, doctor raised smoking, and doctor management, which itself involved several potential outcomes) but also due to the lack of clear presentation of, and signposting to, the results (see below). Abstract The participants are 'past-year smokers' – this is not made clear in the body of the paper, where you describe the number (%) of current, recent, past and never smokers in the total sample. What is the number of the sub-set of patients that you have done your analysis on? Interestingly, the abstract is the only place (apart from the tables) where you state results in terms of ORs (with 95% CIs). I think it would be clearer if you included the key results (in ORs) in the results section as well. Intro Line 10 - You state that previous studies have found that smokers over 65 who quit have reduced mortality; is this also true of those over 70, 75, or 80, or is the evidence lacking? This is key, as if there is little benefit in stopping over a certain age, then patients (and their doctors) may well be justified in not pushing smoking cessation so hard in these age groups. Similarly, in the discussion (p12, line32) you note that NRT appears to be effective in adults over 60 years, but there is limited research in the oldest age groups.
---

Line 19 - Could you add the NICE ref in the first sentence of para 2. Presume it is different to Ref 5?
For consistency, please use either 'age-group' (e.g. p6 line 40 and p7 line 53) or 'age group' (p7 line 8) throughout, but not both.

Method

Please add a sentence about ethics approval and participants consent.

You do mention this in the discussion, but I was struck by your choice of an occupation-based measure of SES, given your exposure of interest is age and many of those in the older age groups will be retired (and therefore put in the lowest SES group, unless they live with younger family). Would it have been possible to use a postcode-based measure such as IMD?

Results

My main issue with the presentation of results is that there is no table describing the included participants with numbers. This makes it hard to interpret the odds ratios that follow. Supplementary Table 1 has column %, but this is far harder for the reader to make an assessment of than numbers and row %.

With regard to signposting, it would be easier to follow if you introduced each table of results in order in the text, e.g. "Table 1 shows..." Table 1 is not mentioned at all in the results section and most of the other tables are only in brackets.

Figure 1 and supplemental Table 2 use different age groups (16-54 is the ref, 55-64, 65-74, 75+) and Tables 3 and supp. Table 3 do not have a 80-84 age group – I presume this is because of smaller numbers for the former (related to NRT and use of cessation services), but this needs to be justified, or at least acknowledged somewhere.

P11, line 3 "...doctors appeared not to offer any support to help quitting". I suggest this over-states your findings. Similarly, throughout the discussion section be wary of phrasing of sentences such as "GPs were less likely to raise smoking with men and less likely to offer any form of support." Remember this is self-reported – perhaps "...men report that their GPs..." would be more accurate. I note that throughout the paper you switch between referring to 'doctors' and 'GPs' – not a big deal, but might be worth sticking with GPs, as that is the terminology used in the survey.

Discussion

Very thorough strengths and limitations section, but could be broken into more than one paragraph for ease of reading.

Conclusion

I would prefer "This large national study has demonstrated that older smokers REPORT BEING less likely to quit..." and "...GPs are reportedly less likely to offer..."

I hope you find this review helpful for strengthening your manuscript.

REVIEWER	Elena Ratschen University of York, UK
REVIEW RETURNED	18-Jul-2017

GENERAL COMMENTS	Thank you for the opportunity to review this interesting and insightful manuscript describing an analysis of Smoking Toolkit data with a focus on older smokers. The paper is very well written, and I only have few minor comments to make, as follows: GENERAL 1. In various places in the manuscript (both in the abstract and main body of the text), the expression of 'referral to NRT' occurs. I believe that this should be replaced by 'prescription of NRT', unless a referral to Stop Smoking Services for NRT provision was intended (in which case this needs to be clarified). 2. Given the international audience of the paper, I feel that providing a somewhat more detailed account of the way smoking cessation advice and treatment are delivered in the UK might be valuable. Although this is implied in the manuscript, it may be difficult to understand the full context for readers unfamiliar with these structures. 3. What is the authors' rationale for restricting the data analysis to NRT prescription, rather than include other evidence-based medications (bupropion/varenicline)? METHODS 1. I wondered whether the potential confounders included a measure of comorbidity at all? If not, the I presume the reason would be that no such data were available? 2. The authors state that the secular period may 'influence the availability of cessation therapies'. Seeing as the study commenced in 2006 and ended in 2015, I am not entirely sure what exactly is meant by this; Stop Smoking Services were commissioned during this time, NICE guidance was published, and pharmacotherapies licensed... 3. Page 9, line 25: 'GP referral for NRT, stop smoking counselling or just advice'. Suggest edit to clarify. Do you mean brief advice by the GP, prescription of NRT, or referral to a Stop Smoking Service (for evidence-based behavioural and pharmacological treatment)? DISCUSSION 1. Page 12, line 3: should this be 'the oldest' rather than 'older participants'? 2. Page 14, line 6: end of this sentence needs a reference.
---

VERSION 1 – AUTHOR RESPONSE

Reviewer 1

1. Could include justification as to how you defined past smoker on the basis of 1 year, as well as the age bands selected.

Response: We have added a statement justifying these issues (page 6).

2. Make clear earlier (under heading - 'dependency and past use of services') why you dichotomised reduction behaviour variables.

Response: We have added a statement to the text (page 6).

3. Explicitly state which variables you entered as independent variables in the statistical analysis.

Response: We have added a statement added to the methods (page 7).

4. If there is any background on the GPs include this as well - for example, GPs gender, age, if sole practitioner, etc.

Response: Unfortunately this data is not collected as part of the STS, so we could not make the requested additions.

5. Implications: there is an appropriate suggestion of the need to do qualitative research. However, there is not a great deal of discussion of implications for policy and practice in direct relation to the authors' data.

Response: We believe that there is a need to gain a better understanding of the underlying demand for and acceptability of services among older adults before making more recommendations.

6. Conclusion: is not saying much - it is mostly just a summary.

Response: The conclusion is short, but we think that it sums up the findings and implications succinctly, and hope the reviewer agrees.

7. There are a few sentences that overuse comas. For example, in line 54 in the abstract.

Response: We have removed extra commas throughout.

Reviewer 2

1. In the abstract, the objective, I wonder whether 'influenced' is the correct term.

Response: We have changed the wording (page 2).

2. In the first sentence of the introduction, I think relative numbers of the smoking problem would make it more comprehensible for the reader.

Response: We have added numbers to the sentence (page 4).

3. In the second paragraph of the introduction, line 20-21, it is mentioned that smoking cessation advice should be provided to all smokers regardless of age including the oldest old. I am wondering what the arguments were. I suggest stating these. Also, it is not clear what is meant with oldest old.

Response: We have removed the wording 'the oldest old' and left the statement at 'regardless of age'. NICE don't state their arguments for not discriminating by age, but we put forward our view within the paper. We have added a statement to the discussion highlighting the evidence for quitting at older ages (page 4).

4. "Advice and support to older smokers may often be missed..." (line 28), I am wondering whether there are any estimates showing how often. The paper referenced discusses attitudes, beliefs of clinicians and knowledge in relation to older smokers and quitting, rather than estimating the frequency of missed opportunities.

Response: This is an interesting point and we are not aware of any evidence on the frequency of missed opportunities.

5. Line 30-31, I miss some elaboration about the potential positive effects of other interventions than NRT. We specifically mentioned NRTs safety because there is some evidence that older smokers have considered it potentially harmful, and there was no such concern evidenced about non-pharmacological interventions.

Response: The quoted paper looks only at pharmacotherapies and NRT was the only one we could identify with any evidence base for older smokers.

6. In the last paragraph of the introduction, line 8, it is not entirely clear what is meant with "far greater detail".

Response: We have elaborated in the text to show that 'far greater detail' refers to the ability to consider age bands in the over 60's rather than amalgamating data (page 5).

7. In the same paragraph and line, authors mentioned several hypotheses while they look like research questions to me. I suggest rephrasing them.

Response: We have rephrased these sentences (page 5).

8. In the methods, line 51, I am wondering whether there were specific reasons for using 5 year bands and 80+ as oldest age-group. If there are reasons, the paper would benefit from more transparency.

Response: We have added a statement to the methods clarifying this (page 6).

9. Page 8, line 3-4, it is not clear how smoking was defined (e.g. not a single puff?).

Response: We added a statement clarifying this as part of the STS (page 6).

10. Page 8, line 10-13, I am wondering whether there is a reference supporting this classification. For instance, the 21 or more cigarettes per day would be classified as heavy smokers? We used the Heaviness of Smoking Index to classify heavy smoking and we have given references to the studies that validated this classification in Ref 18 and 19 (page 6)

11. Page 8, line 25, is there an argument that quit attempts is a good outcome measure for this study? For instance, smoking cessation could also be defined as prolonged abstinence of at least 6 months or even continuous abstinence. Also I suggest adding "in the past 12 months" to clarify what this quit attempts captures. We agree that there are many ways of establishing smoking cessation. However, in the paper our interest is in the way that quitting is approached, and in the support offered. We therefore stand with our use of quit attempts.'

Response: We have added the words 'in the past 12 months' to the text (page 6).

12. In the results section, last paragraph, the results from the sensitivity analyses are not reported properly. I think the paper would benefit from more details. I suggest adding this information in a supplementary table. We have added a supplemental table (Supplemental table 4).

Response: We have not included all the analyses as many models with many factors were compared: the volume of information makes a single table containing the requested details for all factors and models too large for a single table, but have shown the sensitivity results for differences in OR and p values for management by age group (Supplemental file).

13. Page 12, line 11-16, it is discussed that previous studies reported that medical professionals are less likely to advise older patients to stop smoking in a hospital setting and less likely to prescribe NRT or other smoking cessation medications to those over 60 years of age. I am wondering what the reasons are.

Response: We have reported that our empirical findings are in line with other studies, and believe that an investigation into the underlying reasons is very much part of the next phase of work, as indicated in the second paragraph of the discussion (page 10).

14. In the next paragraph, line 20, the low motivation to quit is discussed. I miss some link of findings to motivational interventions that may be provided to this group. What is the role of motivational interviewing for instance?

Response: Unfortunately we cannot make this change as we could only assess interventions reported in the STS and had no data on the role or uptake of motivational interventions.

15. Page 13, line 40, "this is greater for... healthier than the general older population". I am wondering whether there is support for this direction of non-response bias.

Response: We have rephrased the sentence to strengthen the statement that it is an assumption (page 12).

16. It seems to me that this is already described elsewhere but I am wondering whether this paper should report on the research ethics too.

Response: We have added a sentence to the Methods section (page 5).

Reviewer 3

1. Abstract: The participants are 'past-year smokers' – this is not made clear in the body of the paper, where you describe the number (%) of current, recent, past and never smokers in the total sample. What is the number of the sub-set of patients that you have done your analysis on?

Response: We have added numbers to the text so they appear in the body of the paper (page 8) , and they are also shown in supplemental table 2 (Supplemental file).

2. Interestingly, the abstract is the only place (apart from the tables) where you state results in terms of ORs (with 95% CIs). I think it would be clearer if you included the key results (in ORs) in the results section as well.

Response: This is a good point. The text summarises the ORs shown in the tables and we have now added selected ORs to the text within the results section (pages 8 and 9).

3. Intro: Line 10 - You state that previous studies have found that smokers over 65 who quit have reduced mortality; is this also true of those over 70, 75, or 80, or is the evidence lacking? This is key, as if there is little benefit in stopping over a certain age, then patients (and their doctors) may well be justified in not pushing smoking cessation so hard in these age groups. Similarly, in the discussion (p12, line32) you note that NRT appears to be effective in adults over 60 years, but there is limited research in the oldest age groups.

Response: This is important, and we have added a statement and reference to the discussion highlighting the limited evidence (page 11).

4. Line 19 - Could you add the NICE ref in the first sentence of para 2. Presume it is different to Ref 5?

Response: This is ref 5, and we have changed the position of the reference to make this clear (page 4).

5. For consistency, please use either 'age-group' (e.g. p6 line 40 and p7 line 53) or 'age group' (p7 line 8) throughout, but not both. changed age-group to age group throughout and 'Doctor' to GP unless the sentence had already used GP once or was describing "doctor-patient interaction".

Response: We have made the requested changes throughout.

6. Methods: Please add a sentence about ethics approval and participants consent.

Response: We have added a statement to the methods (page 5)

7. You do mention this in the discussion, but I was struck by your choice of an occupation-based measure of SES, given your exposure of interest is age and many of those in the older age groups will be retired (and therefore put in the lowest SES group, unless they live with younger family). Would it have been possible to use a postcode-based measure such as IMD?

Response: We have added a sentence to the strengths and weaknesses section on this (page 11). The anonymised STS data do not include postcode, so an area-based measure of SES cannot be applied. As the reviewer rightly points out, SES is a difficult measure to apply to older adults. The participants in our subset who are over 65 do not all identify as retired (approximately 25% of the over 65s say they are group E.)

8. Results: My main issue with the presentation of results is that there is no table describing the included participants with numbers. This makes it hard to interpret the odds ratios that follow. Supplementary Table 1 has column %, but this is far harder for the reader to make an assessment of than numbers and row %. The numbers were included in Supplemental table 1, but we accept that this was not clear.

Response: We have included numbers in the text to aid clarity (page 8).

9. Figure 1 and supplemental Table 2 use different age groups (16-54 is the ref, 55-64, 65-74, 75+) and Tables 3 and supp. Table 3 do not have a 80-84 age group – I presume this is because of smaller numbers for the former (related to NRT and use of cessation services), but this needs to be justified, or at least acknowledged somewhere.

Response: The reviewer is correct, and we have added acknowledgement to the text (page 8-9) .

10. P11, line 3 “...doctors appeared not to offer any support to help quitting”. I suggest this overstates your findings. Similarly, throughout the discussion section be wary of phrasing of sentences such as “GPs were less likely to raise smoking with men and less likely to offer any form of support.” Remember this is self-reported – perhaps “...men report that their GPs...” would be more accurate.

Response: Thanks you, this is a very valid point, and we have altered the text throughout to emphasise that this was self reported data. We have added the words 'reported' and 'self reported' throughout the discussion, and changed the sentence 'GPs ... were less likely to offer support for older participants' to '... older participants were less likely to report being supported to quit' (page 10). We have changed the sentence 'However, for smokers over 70 years Doctors appeared not to offer any support to help quitting' to 'However, smokers over 70 years were less likely to report that their GPs offered any support to help quitting' (page 9).

11. I note that throughout the paper you switch between referring to 'doctors' and 'GPs' – not a big deal, but might be worth sticking with GPs, as that is the terminology used in the survey.

Response: We have made the requested change throughout.

12. Discussion Very thorough strengths and limitations section, but could be broken into more than one paragraph for ease of reading.

Response: We have made the requested change (page 11-12).

13. Conclusion: I would prefer “This large national study has demonstrated that older smokers REPORT BEING less likely to quit...” and “...GPs are reportedly less likely to offer...”

Response: We have made the requested change (page 10, 11 and 13)

Reviewer 4

1. In various places in the manuscript (both in the abstract and main body of the text), the expression of 'referral to NRT' occurs. I believe that this should be replaced by 'prescription of NRT', unless a referral to Stop Smoking Services for NRT provision was intended (in which case this needs to be clarified).

Response: This is a good point, we have changed the wording to indicate the prescription of NRT by the GP (throughout the paper)

2. Given the international audience of the paper, I feel that providing a somewhat more detailed account of the way smoking cessation advice and treatment are delivered in the UK might be valuable. Although this is implied in the manuscript, it may be difficult to understand the full context for readers unfamiliar with these structures.

Response: We have added a reference to the text in the introduction, referring to the 'type, location and visibility of smoking cessation services' (West et al), which we believe will provide readers with more detailed information should they need it (page 4).

3. . What is the authors' rationale for restricting the data analysis to NRT prescription, rather than include other evidence-based medications (bupropion/varenicline)?

Response: The data are restricted by the STS data collection, which did not include a question about bupropion/varenicline.

4. Methods: I wondered whether the potential confounders included a measure of comorbidity at all? If not, the I presume the reason would be that no such data were available?

Response: The reviewer is correct, we had no available data on comorbidity.

5. The authors state that the secular period may 'influence the availability of cessation therapies'. Seeing as the study commenced in 2006 and ended in 2015, I am not entirely sure what exactly is meant by this; Stop Smoking Services were commissioned during this time, NICE guidance was published, and pharmacotherapies licensed...

Response: We accept this and have added clarification to the text: the statement was made on the basis of personal communications with people working within Smoking cessation services, in which the closure or relocation of some kinds of support groups (referral for counselling) was noted as services were recommissioned (page 7).

6. Page 9, line 25: "GP referral for NRT, stop smoking counselling or just advice'. Suggest edit to clarify. Do you mean brief advice by the GP, prescription of NRT, or referral to a Stop Smoking Service (for evidence-based behavioural and pharmacological treatment)?

Response: We have made the requested changes (page 7).

7. Discussion: Page 12, line 3: should this be 'the oldest' rather than 'older participants'?

Response: The very oldest group in our study is age 80+, we show changes in approach from age 65+, so would prefer to keep the term 'older'.

8. Page 14, line 6: end of this sentence needs a reference. We have added a suitable reference (page 12).

VERSION 2 – REVIEW

REVIEWER	David Blane University of Glasgow, UK
REVIEW RETURNED	11-Sep-2017
GENERAL COMMENTS	The authors have responded adequately to the reviewers' comments.